# Distribution and Prevalence of *Coxiella burnetii* in Animals, Humans, and Ticks in Nigeria: A Systematic Review

Kaka A. Muhammad [1], Usman N. Gadzama [1] and ThankGod E. Onyiche [2,3,*]

1 Department of Biological Sciences, University of Maiduguri, P. M. B. 1069, Maiduguri 600230, Nigeria; kmuhammad5@gmail.com (K.A.M.); gadzamausman69@gmail.com (U.N.G.)

2 Department of Veterinary Parasitology and Entomology, University of Maiduguri, P. M. B. 1069, Maiduguri 600230, Nigeria

3 Department of Biological and Environmental Sciences, Faculty of Natural Sciences, Walter Sisulu University, PBX1, Mthatha 5117, South Africa

* Correspondence: et.onyiche@unimaid.edu.ng or tonyiche@wsu.ac.za

**Abstract:** 'Query' (Q) fever is a neglected but emerging or re-emerging zoonotic disease caused by the bacterium *Coxiella* (C.) *burnetii*. Several host species are considered or speculated to be the primary reservoir hosts for human infection. In the past, several research groups in Nigeria have evaluated the prevalence of *C. burnetii* in various vertebrate and invertebrate hosts. Currently, there is a paucity of knowledge regarding the epidemiology of the pathogen in Nigeria with limited or no attention to control and prevention programs. Therefore, this review was undertaken to comprehend the current situation of *C. burnetii* infection in human, domestic and peri-domestic animals, and some tick species in Nigeria since 1960 with the aim to help identify future research priorities for the country. A comprehensive literature search was performed using the PRISMA guidelines on five scientific databases including Google Scholar, PubMed, AJOL, Science Direct, and Scopus for articles published from Nigeria dealing with the screening of blood, milk, or tick DNA for evidence of *C. burnetii* using any standard diagnostic approach. Of the 33 published articles subjected to full-text evaluation, more than 48% of the articles met the inclusion criteria and were thus included in this review. We observed different ranges of prevalence for *C. burnetii* antibodies from four vertebrate hosts including cattle (2.5–23.5%), sheep (3.8–12.0%), goats (3.1–10.9%), and humans (12.0–61.3%). Additionally, the use of molecular diagnostics revealed that the DNA of *C. burnetii* has been amplified in eight tick species including *Hyalomma* (*Hy*) *dromedarii*, *Hy. truncatum*, *Hy. impeltatum*, *Hy. rufipes*, *Hy. impressum*, *Amblyomma* (*Am.*) *variegatum*, *Rhipicephalus* (*Rh.*) *evertsi evertsi*, and *Rh. annulatus*. Two rodent's species (*Rattus rattus* and *Rattus norvegicus*) in Nigeria were documented to show evidence of the bacterium with the detection of the DNA of *C. burnetii* in these two mammals. In conclusion, this review has provided more insight on the prevalence of *C. burnetii* and its associated host/vector in Nigeria. Domestic animals, peri-domestic animals, and ticks species harbor *C. burnetii* and could be a source of human infections. Due to the paucity of studies from southern Nigeria, we recommend that research groups with interest on vector-borne diseases need to consider more epidemiological studies in the future on *C. burnetii* prevalence in diverse hosts to help unravel their distribution and vector potentials in Nigeria as a whole.

**Keywords:** epidemiology; emerging pathogen; prevalence; *Coxiella burnetii*; Q fever; associated vectors; Nigeria





## 1. Introduction

*Coxiella burnetii* is a deadly bacterium causing Q fever and is considered as an emerging or re-emerging zoonotic pathogen of medical importance caused by the Gram-negative obligate intracellular bacterium; belonging to the phylum Proteobacteria, class Gammaproteobacteria, and family Coxiellaceae, it affects both animals and humans with worldwide

distribution [1–3] and is responsible for Q fever. The first isolate of *C. burnetii* was originally from *Dermacentor andersoni* ticks collected in Montana, USA, and for this reason, ticks are considered as vectors for transmission [4]. Domestic ruminants (sheep, goats, and cattle) have been shown to shed the bacteria in amniotic fluid, vaginal mucus, milk, urine, and feces [4,5]; thus, they are regarded as the main reservoir for human infections [6].

Q fever was listed among the emerging infectious disease by the World Health Organization (WHO), the Food and Agricultural Organization (FAO), the European Food Safety Authority (EFSA), and the Centre for Disease Control and Prevention (CDC) [7,8]. Q fever manifests major clinical symptoms like abortions and stillbirths that lead to significant economic losses in livestock. Usually, Q fever can be asymptomatic in humans, but when the symptom prevails, high fever, severe pneumonia, or hepatitis are some of the common signs of acute infection [8,9]. Endocarditis, vasculitis, lymphadenitis, prosthetic joint arthritis, persistent fatigue, and osteomyelitis are chronic manifestations of the disease, but are rare with fatal effects on patients [9,10].

Ticks are considered to be the main arthropod host and vector of *C. burnetii*, transmitting the pathogen to mammals either through their bite or contamination with their fecal materials [11–13]. The detection of *C. burnetii* in different species of ticks has been reported in Nigeria [14–16] and other regions of the world like Iran, Europe, and Australia [17,18]. Also, domestic animals may play a vital role as reservoir hosts and sources of human infection [19]. Shared grazing pasture and/or water sources in the same pastoral environment by different herds increases the potential of acquiring *C. burnetii* infection [12]. Both animals and humans can also be infected through the inhalation of airborne particles contaminated with *C. burnetii* [11,12,19,20]. The inhalation of infective dose <10 bacterial cells by occupational risk groups including veterinarians and farm workers from infected animals and their products poses a significant risk to the acquisition of *C. burnetii* infection [20,21]. However, peri-domestic animals like rats, cats, rabbits, and dogs could also play an important role in transmitting *C. burnetii*. The screening of rodents by Kamani and co-workers recorded a prevalence of 2.2% in Nigeria [22]. Vanderburg and co-workers also documented their findings on the epidemiology of *C. burnetii* across Africa and concluded with evidence that the pathogen is endemic in cattle, small ruminants, and humans across the continent, with seroprevalence ranging from 11 to 33% in sheep, 13–24% in goats, 4–55% in cattle, and 1–32% in humans [21].

In Nigeria, a high prevalence of 44% was recorded in a sero-epidemiological investigation of Q fever among hospitalized patients [23]. However, the epidemiology of Q fever has not been properly comprehended by medical, veterinary personnel, and animal handlers in the country due to evident neglect of the disease. In the last few decades, a number of individual studies have been published on the prevalence of *C. burnetii* in livestock, humans, and ticks, but no effort has been channeled to consolidate all the published literatures into a single one-stop document for easy reference. Hence, this systematic review was conducted on *C. burnetii* infection among humans, animals, and tick vectors to ascertain the level of their prevalence and distribution within the country with the aim to consolidate the existing body of knowledge and identify gaps for future investigation regarding *C. burnetii* infection in Nigeria.

## 2. Materials and Methods

### 2.1. Search and Selection Criteria

A systematic search was conducted for articles published from January 1960 to September 2022 by two of the co-authors, independently adopting the guidelines of the Preferred Reporting Items for Systematic Reviews and Meta-Analyses (PRISMA) [24]. The literature search was conducted in Google Scholar, PubMed, AJOL, Science Direct, and Scopus databases, using the keywords "Distribution", "Prevalence", "*Coxiella burnetii*", "Sheep", "Goats", "Cattle", "Camel", "Domestic animals" "Peri-domestic animals", "Human", "Tick", "Nigeria". These keywords were used independently or in combination using the Boolean operators "AND" and/or "OR". The search was restricted to articles

reporting the work performed on *C. burnetii* in domestic and peri-domestic animals, and humans as well. Lastly, we included studies investigating the detection of *C. burnetii* DNA in both questing and blood-fed ticks in Nigeria. The search results were screened by removing duplicates and then selecting articles with relevant titles and abstracts. The selected articles were downloaded to enable the screening of the full text for eligibility.

### 2.2. Inclusion and Exclusion Criteria from the Study

Articles were considered valid when they answered our research questions, and the pre-set criteria were put into place to assess the eligibility of every article. The following question was used to screen for the best articles for this study: (i) Is the article written in English, peer-reviewed, and published from January 1960 to September 2022? (ii) Did the study report the distribution and prevalence of *C. burnetii* in ticks, humans, and domestic/peri-domestic animals in Nigeria? (iii) Did the study capture the geographical region where the study was conducted clearly? (iv) Did the study clearly indicate the diagnostic method employed in the experiment? The review excluded: (i) all articles reporting on the distribution and prevalence of *C. burnetii* outside Nigeria; (ii) studies not written in English, not peer-reviewed, and published before 1960; (iii) studies with unclear sample information (i.e., collection area, type of sample, and pathogens) and unclear diagnostic methods employed for the experiment.

### 2.3. Data Extraction and Analysis

All studies that fulfilled the inclusion criteria were documented on a spreadsheet after the data extraction. The following information was considered paramount on the spreadsheet: author names, study design, geographical area, sample collection year, number and sample type screened, diagnostic techniques used for screening, tick species identified if present, and the percentage of *C. burnetii* infection detected from the samples. The data were compiled and analyzed using Excel version 2010.

## 3. Results

### 3.1. Outcome of the Literature Search

A total of 1475 articles were obtained from the systematic search from five databases using the procedure outlined in Figure 1. Of this number, 276 (18.71%) duplicates were excluded at the initial screening, leaving a total of 1199 articles. After screening the titles and abstracts, review papers, and textbooks, 1166 articles were deemed ineligible and were excluded from the review. Thirty-three (33) articles were selected for full-text evaluation, of which 17 articles were removed as their focus was not on *C. burnetii*. Consequently, 16 articles were considered eligible for inclusion and are thus discussed in this review (Figure 1).

### 3.2. Characteristics of the Studies Included in the Systematic Review

The features of all the studies included in this review are summarized in Table 1. The studies included in this review used either an Enzyme-Linked Immunosorbent Assay (ELISA) [3,25–33] or a Microimmunofluorescence test [23] to carry out serological analysis or molecular assays [14–16,22] for the detection of *C. burnetii*. The Capillary Agglutination Test (CAT) was also employed to screen the milk samples for *C. burnetii* [34]. Most of the reported studies screened multiple animal hosts, with a handful of studies involving a single host (Table 1). In total, five host species including cattle, sheep, goats, rodents, and humans were the center point of this study, while ticks were the only invertebrate host studied. Domestic animals (cattle, sheep, and goats) were the most studied, and *C. burnetii* DNA was reported in 4 out of 169 (2.4%) rodents, comprising of 3 out of 121 (2.5%) *Rattus norvegicus* and 1 out of 48 (2.1%) *Rattus rattus* screened (Table 1). All the samples collected from all the investigated animals originated from both southern and northern Nigeria (Figure 2). Most of the eligible studies were from domestic animals across the northern part of Nigeria comprising of fourteen studies, while only two studies were conducted in the

southern part of the country (Table 1). The breakdown of figures of the total number of domestic animals sampled (cattle, sheep, and goats) in the northern region between 2014 and 2021 includes cattle (n = 1673), sheep (n = 1330), and goats (n = 1042). With regards to ticks, eight species have so far been screened for the detection of *C. burnetii*, including *Hy. truncatum*, *Am. variegatum*, *Rh. evertsi evertsi*, *Hy. dromedarii*, *Hy. rufipes*, *Hy. impeltatum*, *Hy. Impressum*, and *Rh. (Boophilus) annulatus.* Only one study screened rodents for *C. burnetii* DNA. Lastly, 169 milk samples were screened using the capillary agglutination test (CAT) for positivity to detect *C. burnetii* in the north region of Nigeria.

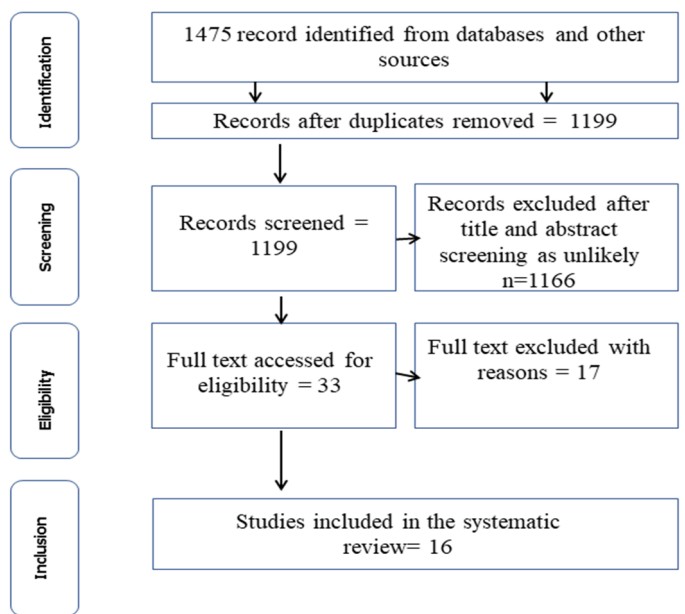

**Figure 1.** PRISMA flow chart showing the systematic selection for inclusion and exclusion of articles in this study.

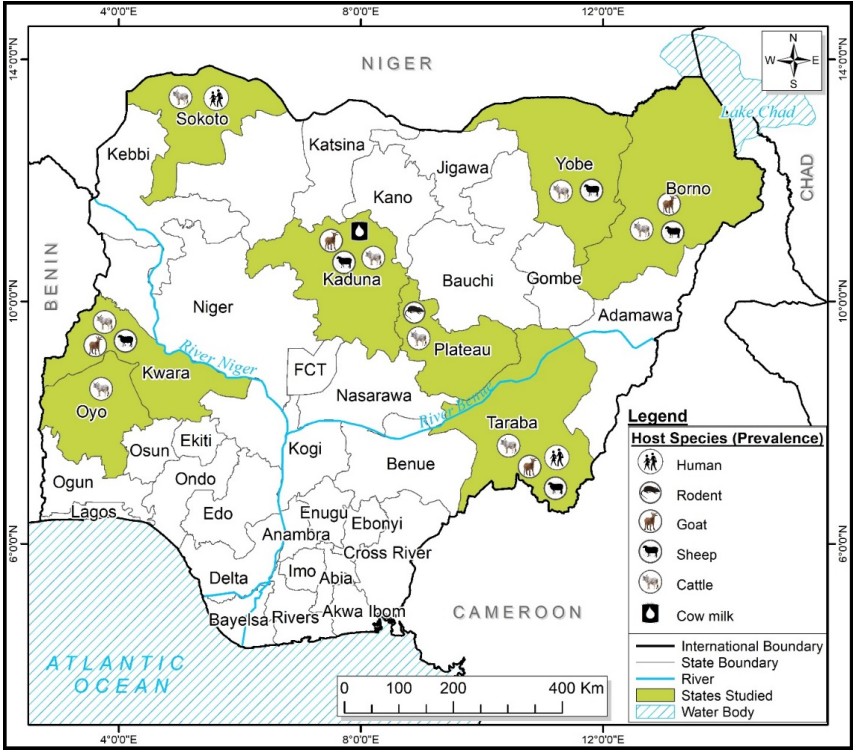

**Figure 2.** Map of Nigeria indicating areas where these studies were conducted.

**Table 1.** Characteristics of all eligible studies reporting the occurrence of *Coxiella burnetii* in different hosts and milk samples in Nigeria.

| Table Cont. | Reference ID | Study Design | Region (States) | Diagnostic Technique | Total AniSpecimen | Host Species (Prevalence) | Presence of Ticks (Infection Rate) | Tick Species |
|---|---|---|---|---|---|---|---|---|
| 1 | Elelu et al. [3] | Cross-sectional | North (Kwara, Plateau and Borno) | ELISA | 538 | Cattle (28/268; (10.44%) Sheep (1/26; 3.8%) Goat (5/158; 3.16%) | - | - |
| 2 | Onyiche et al. [16] | Cross-sectional | North (Kano, Jigawa and Sokoto) | PCR | 176 | Camel | Yes (17/593; 2.9%) | *Hyalomma truncatum, Amblyomma variegatum, Rh. evertsi evertsi, Hyalomma. dromedarii, Hy. rufipes, Hy. impeltatum, Hy. Impressum* |
| 3 | Kamani et al. [22] | Cross-sectional | North (Plateau) | PCR | 194 | Rodent; 4/194 (2.2%) | NA | - |
| 4 | Adamu et al. [27] | Cross-sectional | North (Kaduna) | ELISA | 400 | Cattle (25/400; 6.2%) | NA | - |
| 5 | Adamu et al. [28] | Cross-sectional | North (Kaduna) | indirect enzyme-linked immunosorbent assay ELISA | 400 | Goats (35/400; 8.7%) | NA | - |
| 6 | Adamu et al. [26] | Cross-sectional | North (Borno) | ELISA | 768 | Sheep (46/384; 11.9%) Goats (42/384; 10.9%) | NA | - |
| 7 | Reye et al. [14] | Cross-sectional | South (Oyo) | PCR | 836 | - | YES (19/136; 14%) | *Amblyomma variegatum, Rh. (Boophilus) annulatus, Hyalomma impeltatum, Rhipicephalus evertsi* |
| 8 | Ogo et al. [15] | Cross-sectional | North (Plateau and Nasarrawa) | PCR | 40 | - | YES (10/40; 25%) | *Amblyomma. Variegatum* |

| Table Cont. | Reference ID | Study Design | Region (States) | Diagnostic Technique | Total AniSpecimen | Host Species (Prevalence) | Presence of Ticks (Infection Rate) | Tick Species |
|---|---|---|---|---|---|---|---|---|
| 9 | Tukur et al. [31] | Cross sectional | North (Kaduna) | ELISA | 539 | Cattle: 78/539; (14.5%) | NA | - |
| 10 | Cadmus et al. [32] | Cross-sectional | North (Sokoto) | ELISA | 503 | Human: 84/137; (61.31%) Cattle: 9/366; (2.45%) | NA | - |
| 11 | Nyifi et al. [25] | Cross-sectional | North (Taraba) | ELISA | 350 | Human (6/50; 12%) Goat (10/100; 10%) Sheep (9/100; 9.0%) Cattle (13/100; 13%) | NA | - |
| 12 | Blondeau et al. [23] | Case control | North (Sokoto) | Microimmunofluorescence test | 75 | Human (33/75; 44%) | NA | - |
| 13 | Adamu et al. [29] | Cross-sectional | North (Kaduna) | ELISA | 400 | Sheep (32/400; 8%) | NA | - |
| 14 | Cadmus et al. [33] | Cross-sectional | South (Oyo) | ELISA AND RBPT | 149 | Cattle (35/149; 23.5%) | NA | - |
| 15 | Adamu et al. [30] | Cross-sectional | North (Yobe) | ELISA | 420 | Sheep (49/420; 11.7%) | NA | |
| **Study on Milk Samples** | | | | | | | | |
| S/N | Reference ID | Study Design | Region (States) | Diagnostic Technique | Total Animals Screened | Host Species (Prevalence) | Infection rate | |
| 1 | Adesiyun et al. [34] | Cross-sectional | North (Kaduna) | Capillary Agglutination Test (CAT) | 169 | Cow milk (41/169; 24.2%) | NA | |

On a general note, fewer studies have so far been carried out in the southern part of Nigeria compared to the northern region. Only one study reported on domestic animals (cattle) with 149 samples using the ELISA diagnostic technique. Similarly, only one study in southern Nigeria screened ticks' DNA for *C. burnetii* compared to two studies in the north. However, across Nigeria, the prevalence of *C. burnetii* in cattle ranges from 2.5 to 23.5% [3,25,31–33]; for sheep, it ranges from 3.8 to 12.0% [3,26]; and for goats, from 3.1 to 10.9% [3,25,31]. The result for human-reported prevalence ranges are from 12.0 to 61.3% [25,32].

### 3.3. Host-Vector Relationships

*Coxiella burnetii* were found to infect several host ranges in Nigeria irrespective of the region or location. The pathogen was reported in several hosts such as cattle, goats, sheep, rodents, and their products, like milk. Only one study screened questing (unfed) ticks collected from vegetation, and this study was conducted in southern Nigeria while feeding ticks was the main emphasis from the studies conducted in the north.

### 3.4. Vector-Pathogen Relationships

*Coxiella burnetii* was documented in *Hy. dromedarii* (3.4%) and *Hy. truncatum* (1.1%) tick pools in northwest Nigeria [16]. Another study from the Plateau and Nasarawa states reported the detection of this pathogen in partially fed *Am. variegatum* ticks [15]. Finally, both *Hy. impeltatum* (1.4%) and *Rh. evertsi evertsi* (2.2%) were reported to harbor *C. burnetii* in southern Nigeria [14].

### 3.5. Diagnostic Assays Employed

All studies involving domestic and peri-domestic animals and humans employed the Enzyme-Linked Immunosorbent Assay (ELISA) as the diagnostic method of choice (Table 1) to detect *C. burnetii* antibodies in animals and human serum. The capillary agglutination test was employed to detect this bacterium from dairy milk where a positivity of 24.2% was reported [34].

## 4. Discussion

This systematic review shows that *Coxiella burnetii* infects a diverse range of animal hosts in Nigeria including cattle, sheep, goats, rodent, and human, as well as invertebrates like ticks.

### 4.1. Diagnostic Techniques Employed So Far in Nigeria

A combination of several diagnostic methods has been employed so far, including the capillary agglutination test (CAT) [34], the Enzyme-Linked Immunosorbent Assay (ELISA) [3,25–33], and PCR [14–16,22] for the study of coxiellosis in humans, animals, milk, and tick vectors in the country. However, the studies involving tick species employed morphological identification of tick species and semi-nested PCR, which was used to detect the genetic materials of *C. burnetii* from the vectors targeting the *16*S rDNA gene [16]. Other studies targeted the *htpB* gene for the detection of *C. burnetii* [8–10,14].

The majority of the studies (n = 10) adopted ELISA, a serological technique, as the diagnostic procedure of choice, while four (4) studies adopted the PCR assay to detect *C. burnetii*. The phase II antigen was also a target for the ELISA assay to detect *C. burnetii* [32]. Nonetheless, PCR has been regarded as the best for detecting *C. burnetii*, but this is still out of the reach for most scientists in sub-Saharan Africa including Nigeria due to the cost involved in the purchase of consumables and equipment. Thus, this could be the reason for more scientists going for ELISA, which is cost-effective. There is a need to adopt PCR for scientific research because it is an excellent technique for the rapid detection of pathogens, including those difficult to culture. It also has the capacity to generate both qualitative and quantitative results from an experiment [35].

*4.2. Ticks as Vector of Coxiella burnetii*

Ticks have been identified as a potential risk for coxiellosis in domestic animals and livestock [12]. In wild animals, ticks may play a significant role as reservoirs of *C. burnetii* [36].

A further credence to the potential role of ticks as vector of *C. burnetii* is the isolation of this bacterium in over 14 soft tick species and 40 hard ticks species collected from domestic, wild animals, and vegetation [4,37]. The excretion of infectious feces by ticks containing up to $10^{10}$ organisms per gram of feces emphasizes the potential risk of tick-borne infection posed by tick excreta [12].

Ticks are considered the most important vector of pathogens, including as a vector of *C. burnetii*, as they maintain the infection in domestic animals [8,37]. Transmission may occur through a tick bite or exposure to the infected excreta expelled by ticks onto the skin of the animal host or environment [12]. Across the country, empirical evidence indicates that *C. burnetii* has been detected in questing ticks, that are in the process of host seeking, and feeding ticks, which are already on their host [14–16]. In the study by Reye et al. [14], in southwestern Nigeria, exactly four species of ticks, namely *Am. Variegatum, Rh. Annulatus, Hy. Impeltatum,* and *Rh. Evertsi*, were recorded to harbor the DNA of *C. burnetii* with an overall prevalence of 14.0%. Of the four tick species, *Am. Variegatum* had the highest prevalence of 33.3% to *C. burnetii*, which is comparable to the result obtained at the Niakhar region of Senegal [38], where the infection rate was 37.6%. Furthermore, *Am. variegatum* ticks has been recorded in a handful of previously documented works accounting for the bulk of the infection among other screened ticks, as observed in Ghana [39], western Kenya [40], and north-central Nigeria [15]. Other species of ticks that have been documented in Nigeria to harbor *C. burnetii* includes *Hy. Truncatum* and *Hy. dromedarii* with a minimum infection rate (MIR) of 2.9% [16]. This finding was similar to the result from Korea where a prevalence of 1.2% and 1.61% was reported from *Haemophysalis* (*Haem*) *longicornis* and *Haem. flava* ticks, respectively [41]. In a nutshell, the findings from different studies suggested that *Am. Variegatum* [14,15,42] and *R. evertsi evertsi* [14,38] ticks are potential vectors of *C. burnetii* and domestic animals may play a vital role as reservoir hosts and sources of human infection [19]. However, transmission experiments are needed to confirm the vectorial role of these tick species. Considering the relationship between ticks and, more especially, free-range grazing animals by herdsmen, the potential tendency of acquiring *C. burnetii* infection from ticks could be high during the grazing of animals in the pastoral environment, as suggested by Koka et al. [37]. In fact, the experimental transmission of *C. burnetii* have been reported in several species of ticks, where the bacteria were found to multiply in the mid-gut of infected ticks and the pathogen was expelled into the environment or onto the skin of the animals via feces [43].

*4.3. Coxiella burnetii in Domestic and Peri-Domestic Animals*

Milk is regarded as the most common route of *C. burnetii* shedding in cattle and goats [44–46].

Domestic animals such as cattle, sheep, and goats serve as the primary reservoir hosts and sources of *C. burnetii* infection. In animals, infections are mainly asymptomatic, but stillbirth, abortion, the delivery of weak offspring, and infertility are reported to occur [4,16]. Across Nigeria, the prevalence ranges are from 2.5 to 23.5% in cattle [3,25,31–33]. Moreso, milk in dairy cows can also be a potential source of *C. burnetii* transmission, as evidence shows that this pathogen has been reported from the milk collected from cattle raised under two management systems (semi-intensive and Fulani nomadic) in Nigeria [34]. Of the 162 milk samples screened for *C. burnetii* using PCR, the bacteria was reported in 10.2% of goats, 18.6% in sheep, and 15% in cattle milk in Iran [47]. Furthermore, three studies conducted in Italian cattle herds in 2013 and 2014 also reported differences in *C. burnetii* prevalence of 40.0% and 60.0%, respectively [48]. All these observations point to the role of milk as a vehicle in the transmission of *C. burnetii* to humans. In our study, we also observed that among small ruminants, the infection rates for sheep was

comparatively higher at 3.8–12.0% [3,26], compared to goats at 3.1–10.9% [3,25,31,32]. On the contrary, the infection rate in a similar study in Kenya showed that the infection was higher in goats (20.0–46.0%) compared to sheep (6.7–20.0%), while the results for other host includes cattle (7.4–51.0%) and camels (20.0–46.0%) [49]. Furthermore, Nahed et al. [50] reported the seropositivity of *C. burnetii* from three governorates surrounding Cairo, Egypt, as follows with 13% prevalence in cattle, 23% in goats, and 33% in sheep. Similarly, in Chad Republic, an 80.0% prevalence in camels, 4.0% in cattle, 13.0% in goats, and 11.0% in sheep has been recorded [51]. Peri-domestic animals like rats, cats, rabbits, and dogs can also play an important role in the transmission of *C. burnetii.* The screening of four species of rodents (*Rattus rattus, Rattus norvegicus, Mus musculus,* and *Cricetomys gambiances*) by Kamani et al. [22] registered a prevalence of 2.2% in Nigeria. A higher prevalence of 45.0% has been documented in Zambia after screening three species of rodents (*Saccostonus campestris*, Gerbillinae spp, *and Mastomys natalensis)* [52]. The finding from Heixiazi Island in China showed a prevalence of 18.0% [53], which is similar to the finding in Kenya where a prevalence of 13.0% was reported [49]. However, rodents do not seem to automatically play a significant role in the maintenance of *C. burnetii* infection; rather, they represent accidental or dead-end hosts [54]. The rodent has more chances to be infected with *C. burnetii* as larger rodents became mature and move widely, with a high tendency of being a reservoir host from which domestic animals, especially cats, which are natural predators of these animals, may become infected [19,53].

### 4.4. Coxiella burnetii in Humans

Research regarding Q fever in humans has largely increased during the past decade. In this current study, we identified some eligible articles that screened humans for *C. burnetii* in Nigeria and reported a prevalence of up to 61.3% after screening 137 pastoralists in Sokoto State [32] and, in the same vein, a prevalence of 12.0% was also reported in Jalingo, Taraba State [25]. All studies on humans were undertaken in the northern part of Nigeria. Outside Nigeria, a handful of studies have reported the occurrence of *C. burnetii* infection. For instance, Schelling et al. [51] reported a seroprevalence in humans at 1.0%, while a seroprevalence of 32.0% was registered in a Nile Delta village in Egypt [55]. A report from western Kenya also revealed an overall seroprevalence of 2.5% of *C. burnetii*, which was higher among males (3.3%) and children aged 5–14 (2.2%), compared to females (1.9%) in an age range between 15 and 24 years (2.2%) [40]. In Niamey, Niger Republic, 10.0% of children aged 1 month to 5 years were seropositive, and in Ghana's rural Ashanti Region, 17.0% of two-year-olds were seropositive [56]. Recent reports across 24 African countries showed the mean seroprevalence for *C. burnetii* in humans was 16.0%, while molecular detection showed 3.0% [57]. The high prevalence recorded may be due to local management practices, environmental and climatic conditions, the methods of screening samples, and geographical variations [31]. Outside Africa, Cetinkol et al. [58] reported a positivity rate of 28.6% among veterinarians in Hatay, 26.0% in Istanbul, up to 80.0% in veterinary health technicians, and 33.3% from veterinary students in Turkey, which correlate with the findings from Nigeria [32] and Egypt [50]. A similar report from France showed a higher seroprevalence of 88.9% among livestock handlers and veterinarians than the general population [59]. In most cases, the *C. burnetii* infection remains asymptomatic or presents as non-specific flu. Thus, it remains undiagnosed [19].

### 4.5. Coxiella burnetii Infection in Northern Nigeria

The majority of the published studies on *C. burnetii* infection in humans, animals, and tick vectors were observed to have emanated from the northern region of the country with prevalence ranging from 3.8 to 11.9% [3,25–32]. The present study identified two PCR-based studies that examined the presence of *C. burnetii* DNA in ticks in the northern part of Nigeria. Onyiche et al. [16] screened seven species of ticks, but only two species had *C. burnetii* genetic material with an infection rate of 2.7%, while Ogo et al. [15] reported a prevalence of 25.0% from one species of tick. However, two human-based studies were identified with

varying prevalence [25,32]. Cadmus et al. [32] reported 61.3% seroprevalence for *C. burnetii* from 137 individuals screened from Sokoto, while 12.0% seroprevalence was reported in Taraba [25]. A similar finding from the studies conducted outside Nigeria in Tana and Garissa in Kenya indicated a prevalence of 24.44% for *C. burnetii* from 974 individuals screened [60], while a lower prevalence of 2.3% was reported in western Kenya [61].

### 4.6. Coxiella burnetii Infection in Southern Nigeria

In southern Nigeria, only two studies were identified pertaining to *C. burnetii* in that region. Both feeding and questing ticks were screened for *C. burnetii* DNA in one of the studies that were conducted in southwest Nigeria and revealed a 14.0% prevalence for *C. burnetii* in ticks [14]. The second study conducted in southwest Nigeria on the seroprevalence of *C. burnetii* in cattle revealed a prevalence of 23.5% [33]. This prevalence is higher than the reported mean seroprevalence for *C. burnetii* in cattle in Africa, which currently stands at 14.0% while PCR detection was 9.0% [57]. On a similar note, a study conducted in some parts of Africa indicated different seropositivity to *C. burnetii*; as 29.9% was registered in dairy cattle in Sudan [62], and in Ethiopia, a seropositivity of 8.2% and 4.7% from semi-intensive and intensive management systems respectively was recorded [63]. Wardrop et al. [61] reported an overall seroprevalence of 10.5% to *C. burnetii* in Western Kenya. However, there are limited studies in southern Nigeria compared to the north. This may be due to several reasons including the higher livestock population and mass movement of animals for grazing in the north compared to the south, and the research interest from academics in the two regions.

### 5. Conclusions

From the present study, it can be concluded that *C. burnetii* infects a wide range of host including humans in Nigeria. The pathogen has been reported in several animal species such as cattle, sheep, goats, rodents, as well as from milk samples and some tick species as potential vectors of *C. burnetii* in Nigeria. As a whole, several diagnostic techniques were employed to detect *C. burnetii*, including molecular detection (PCR) and serological assays (Enzyme-Linked Immunosorbent Assay (ELISA) and Capillary Agglutination Test (CAT)). Hence, this study recommends future research with a focus particularly on domestic animals such as pets, camels, and their products, like milk and meats, for the detection of *C. burnetii,* as this could pave the way for a better understanding of the epidemiology of the pathogens in Nigeria. More so, there is a need for physicians in Nigeria to request Q fever tests in cases of atypical pneumonia in risk groups such as animal handlers, veterinarians, and slaughter workers in case of infection. These occupations have been associated with the increased risk of *C. burnetii* infection globally [26,64].

**Author Contributions:** Conceptualization, K.A.M. and T.E.O.; methodology, K.A.M. and T.E.O.; validation, T.E.O.; formal analysis, K.A.M. and T.E.O.; investigation, K.A.M. and T.E.O.; writing—original draft preparation, K.A.M.; writing—review and editing, T.E.O. and U.N.G.; supervision, U.N.G. and T.E.O. All authors have read and agreed to the published version of the manuscript.

**Funding:** This research received no external funding.

**Institutional Review Board Statement:** Not applicable.

**Informed Consent Statement:** Not applicable.

**Data Availability Statement:** Not applicable.

**Conflicts of Interest:** The authors declare no conflict of interest.

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
