# Peer review of "Distribution and Prevalence of Coxiella burnetii in Animals, Humans, and Ticks in Nigeria: A Systematic Review"

_2036-7449, doi:10.3390/idr15050056_

Round 1
Reviewer 1 Report
The review topic is of interest and fulfills a need; however, there are major concerns that limit the usefulness of the review in its current form. These comments should be addressed in order to ensure that the information being provided in the review is accurate and supported by published data. There is some text that matches too closely to the referenced studies and should be paraphrased to avoid potential plagiarism.
Major comments
Abstract
- Lines 28-29: It is not clear what the percentages are. Please clarify.
Introduction
- Line 58: ‘It is asymptomatic in humans’ – Q fever can be asymptomatic, but is not always asymptomatic as evidenced by the symptoms listed in the rest of the sentence.
- Line 61: Reference #10 is not the appropriate reference for this statement.
- Lines 62-63: Please provide references to support the statement that ticks transmit C. burnetii through bite or contamination with fecal material or remove the statement. The reference #8 does not support this claim. Furthermore, Coxiella-like endosymbionts that possess the IS1111 insertion sequence as prevalent in ticks (Reference 10 and Duron, FEMS Microbiology Letters 2015).
- Lines 67-68: Please provide references to support the statement that grazing increases the potential for livestock to acquire C. burnetii infection from ticks. Reference #17 does not support this conclusion. It simply detected IS1111 insertion sequences in DNA from ticks. Please refer to the comment above. If references cannot be provided that support this statement, please remove it.
- Reference 10 should be discussed in more detail and studies with PCR-based analysis of ticks should be interpreted with caution.
- Line 68-69: This statement is not supported by reference #10. Please provide appropriate references.
- Lines 71-75: Reference 19 documents C. burnetii seropositivity as 3.6% in cattle herds not 4-55% and no mention of sheep and humans. Please correct the reference to support the statement.
- Lines 75-75: Reference 19 does not support this statement.
- Lines 77-78: Reference 20 does not support this statement. The statement is almost verbatim from a line in Reference 20. The authors should be cautious of this to avoid potential plagiarism and the original work is not properly cited.
Materials and Methods
- Line 90: Please clarify why January 1960 was used as the starting date for inclusion.
- Line 98: Please clarify why milk samples were selected for inclusion over other sample types.
- Line 111: Please clarify what is meant by ‘Did the study capture area of study clearly?’
Results
- Line 131 and Fig 1: This reads as though peer reviewed articles were removed as ineligible. Please clarify.
- Lines 133-135: Why was the article in the reference lists not found through the literature search? Could the search have been improved to capture this and similar articles? This should be discussed.
- Table 1:
o This table should separate the information derived from blood samples and milk samples for clarity.
o ‘Total animals/humans screened’ should read total specimens since ticks are included also.
- Line 171-176: More information regarding the methodology in these studies should be included as the htpB sequencing in reference 12 is superior to IS1111 PCR-based methods for detection of C. burnetii and should be discussed in greater detail in the discussion section 4.1
Discussion
- Line 196: Ticks as a vector of Coxiella burnetii – see statement above.
- Lines 197-199: This is an overstatement and should be corrected. Please include a reference that demonstrates that ticks ‘maintain the infection in domestic animals’ or remove the statement. Similarly ‘transmission occurs through a tick bite or exposure…’ this is inaccurate.
- Line 226; This reference does not support the statement.
- Lines 227-233: Should discuss why detection of C. burnetii in milk is important, such as the potential for transmission through consumption of C. burnetii contaminated milk and include relevant references.
- Line 312: Please expand on this. What are the disparities in livestock populations between the two regions?
- Section 4.6: Recommend moving this section to the beginning of the discussion and including details for what types of PCR are employed. What types of ELISA (ie in house vs purchased kits, etc..). Did serology detect phase I and phase II antigens?
Conclusion
- Line 339: Recommend including that these occupations have been associated increased risk of C. burnetii worldwide and include relevant references.
Minor comments
Abstract
- Line 29: Suggest editing ‘From molecular perspective’ to From a molecular perspective’.
- Lines 29-34: Recommend breaking up the long sentence for better clarity.
Introduction
- Line 48: isolates to isolate
- Lines 54 and 62: Paragraph indents should be consistent
- Line 65: Parenthesis in front of references should be removed.
Materials and Methods
- Line 108: should read ‘are the articles’ OR ‘is the article’
- Lines 109-111: Was it a requirement to report on all types of animals in the same study? If not please clarify the sentence.
- Line 112-113: Recommend removing the phrase ‘Never the less’
Results
- Line 120: initials should be initial
- Line 144: cattle sheep and goats should be separated with commas
- Line 166: ‘infect a wide range’
Discussion
- Lines 232-233: ‘comparatively higher compared’ is redundant
- Line 264: ‘Quiet a number of seroprevalence in human’ suggest removing
Conclusion
- Line 329: ‘Infects a wide rang…’
Please review the minor comments for improvements to the language of the paper. This review may benefit from assistance with English language editing.
Author Response
Reviewer 1
The review topic is of interest and fulfills a need; however, there are major concerns that limit the usefulness of the review in its current form. These comments should be addressed in order to ensure that the information being provided in the review is accurate and supported by published data. There is some text that matches too closely to the referenced studies and should be paraphrased to avoid potential plagiarism.
Major comments
Abstract
- Lines 28-29: It is not clear what the percentages are. Please clarify.
Response: This is the range of prevalence documented from published studies against each of the vertebrate host species
Introduction
- Line 58: ‘It is asymptomatic in humans’ – Q fever can be asymptomatic, but is not always asymptomatic as evidenced by the symptoms listed in the rest of the sentence.
Response: Usually, Q fever can be asymptomatic in humans but when the symptoms are apparent, high fever, severe pneumonia, or hepatitis are some of the common signs of acute infection
- Line 61: Reference #10 is not the appropriate reference for this statement.
Response: Reference #10 has now been substituted.
- Lines 62-63: Please provide references to support the statement that ticks transmit C. burnetii through bite or contamination with fecal material or remove the statement. The reference #8 does not support this claim. Furthermore, Coxiella-like endosymbionts that possess the IS1111 insertion sequence as prevalent in ticks (Reference 10 and Duron, FEMS Microbiology Letters 2015).
Response: Thank you for the observation. We have now provided three valid references that support the statement.
- Lines 67-68: Please provide references to support the statement that grazing increases the potential for livestock to acquire C. burnetii infection from ticks. Reference #17 does not support this conclusion. It simply detected IS1111 insertion sequences in DNA from ticks. Please refer to the comment above. If references cannot be provided that support this statement, please remove it.
Response: thank you for the comment. We have now provided a valid reference to support the statement.
- Reference 10 should be discussed in more detail and studies with PCR-based analysis of ticks should be interpreted with caution.
Response: Thank you for the suggestion. Generally, we are aware of the implication of just detecting DNA of C. burnetii in ticks, as this does not imply the competence of the tick to transmit. Consequently, all results that were discussed was done so with caution.
- Line 68-69: This statement is not supported by reference #10. Please provide appropriate references.
Response: We have now provided valid series of references to support the statement.
- Lines 71-75: Reference 19 documents C. burnetii seropositivity as 3.6% in cattle herds not 4-55% and no mention of sheep and humans. Please correct the reference to support the statement.
Response: Vanderburg et al., [19] described the findings on the epidemiology of C. burnetii across Africa with evidence of endemicity in cattle, small ruminants, and humans across the continent, with seroprevalence ranging from 7 % - 33 % in sheep, 4 % - 55 % in cattle, and 1 % - 32 % in humans. Variations in seroprevalence have also been observed within relatively small distances [19].
- Lines 75-75: Reference 19 does not support this statement.
Response: an erroneous reference was initially cited but this has now been corrected to Vanderburg et al., (2014) not Kamga-Walajo 2010.
- Lines 77-78: Reference 20 does not support this statement. The statement is almost verbatim from a line in Reference 20. The authors should be cautious of this to avoid potential plagiarism and the original work is not properly cited.
Response: We acknowledge the error of wrong citation and for direct lifting of sentence. The sentence has now been revised accordingly. Thank you
Materials and Methods
- Line 90: Please clarify why January 1960 was used as the starting date for inclusion.
Response: Thank you for the comment. In systematic review, literature on a given topic over a stated period of time is synthesized to determine the pattern of disease or tredn only if they meet the eligibility criteria. Consequently, in our study, we considered all literature published on Q fever/C. burnetii infection from 1960 to date that are available on the public repository/databases.
- Line 98: Please clarify why milk samples were selected for inclusion over other sample types.
Response: The aim of our study was to determine the prevalence and distribution of C. burnetii in humans, animals and tick vectors in Nigeria. In other words, we were interested on a study that have screened for C. burnetii/Q fever in Nigeria irrespective of sample type. Lastly, we had no specific preference for any sample.
- Line 111: Please clarify what is meant by ‘Did the study capture area of study clearly?’
Response: The major focus of our study was Nigeria and any eligible study for inclusion is expected to clearly state which state/province the study was undertaken in Nigeria. This information is necessary to determine the spatial distribution of C. burnetii in the country.
Results
- Line 131 and Fig 1: This reads as though peer reviewed articles were removed as ineligible. Please clarify.
Response: Our emphasis was to show that all peer-reviewed articles that met the inclusion criteria was eligible for inclusion and if otherwise, were excluded.
- Lines 133-135: Why was the article in the reference lists not found through the literature search? Could the search have been improved to capture this and similar articles? This should be discussed.
Response: Some journals especially the indigenous ones in Nigeria are not well indexed on relevant electronic databases. Hence, during electronic searches, there werel not be picked up. Some of these articles were picked up by looking at the reference list of published articles that were properly indexed on good journals.
- Table 1:
o This table should separate the information derived from blood samples and milk samples for clarity.
Response: According to the PRISMA guideline for systematic review and meta-analysis, the characteristics of all eligible studies included in the review should be displayed on a table. Also, since only one study focused on milk, it is best if all the characteristics of all eligible studies are displayed on a single table. However, we have displayed the results for that involving milk separately.
- ‘Total animals/humans screened’ should read total specimens since ticks are included also.
Response: Now corrected based on the suggestion provided.
- Line 171-176: More information regarding the methodology in these studies should be included as the htpB sequencing in reference 12 is superior to IS1111 PCR-based methods for detection of C. burnetii and should be discussed in greater detail in the discussion section 4.1
Response: Thank you for the suggestion. However, this is a systematic review and we are only synthesizing results obtained from several study. It is beyond the scope of this study to go further to discuss the superiority of a particular gene target over another.
Discussion
- Line 196: Ticks as a vector of Coxiella burnetii – see statement above.
Response: However, studies involving tick species, employed morphological identification of tick species where three different gene were targeted, (12S rRNA, 16S rRNA and cox1 [13] and Semi-nested PCR, was used to detect genetic materials of C. burnetii from the vectors targeting gene 16S rDNA [13}. Other studies used species-specific primers to detect C. burnetii targeting gene htpB [8-11].
- Lines 197-199: This is an overstatement and should be corrected. Please include a reference that demonstrates that ticks ‘maintain the infection in domestic animals’ or remove the statement. Similarly ‘transmission occurs through a tick bite or exposure…’ this is inaccurate.
Response: Sentence have been paraphrased
- Line 226; This reference does not support the statement.
ANS; In fact, experimental transmission of C. burnetii have been reported in several species of ticks, that the bacteria were found to multiply in the mid-gut of infected ticks, where they expelled the pathogen to the environment, or skin of the animals via feces [16].
- Lines 227-233: Should discuss why detection of C. burnetii in milk is important, such as the potential for transmission through consumption of C. burnetii contaminated milk and include relevant references.
Response: We have now further discussed the results on the role of milk in the transmission of C. burnetii.
- Line 312: Please expand on this. What are the disparities in livestock populations between the two regions?
Response: We have now revised the sentence to shed more insight on the disparities in livestock population between the north and south of Nigeria
- Section 4.6: Recommend moving this section to the beginning of the discussion and including details for what types of PCR are employed. What types of ELISA (ie in house vs purchased kits, etc..). Did serology detect phase I and phase II antigens?
Response: The suggested correction has now been effected.
Conclusion
- Line 339: Recommend including that these occupations have been associated increased risk of C. burnetii worldwide and include relevant references.
Response: We have now added the suggested recommendation as well as provide new citations to back up the text. [23, Blut et al. 2013; Bwatota et al. 2022]
Minor comments
Abstract
- Line 29: Suggest editing ‘From molecular perspective’ to From a molecular perspective’.
Response: ‘From a molecular perspective’ correction effected
- Lines 29-34: Recommend breaking up the long sentence for better clarity.
Response: Sentence broken up.
Introduction
- Line 48: isolates to isolate
Response: Corrected as suggested
- Lines 54 and 62: Paragraph indents should be consistent
Response: Corrected as suggested
- Line 65: Parenthesis in front of references should be removed.
Response: Corrected as suggested
Materials and Methods
- Line 108: should read ‘are the articles’ OR ‘is the article’
Response: Corrected as suggested ‘is the article’
- Lines 109-111: Was it a requirement to report on all types of animals in the same study? If not please clarify the sentence.
Response: Yes, our study focuses on all types of animals.
- Line 112-113: Recommend removing the phrase ‘Never the less’
Response: Removed ‘Never the less’ as suggested
Results
- Line 120: initials should be initial
Response: ‘ initials’ corrected to ‘initial’
- Line 144: cattle sheep and goats should be separated with commas
Response: commas inserted.
- Line 166: ‘infect a wide range’
Response: Correction effected.
Discussion
- Lines 232-233: ‘comparatively higher compared’ is redundant
Response: Sentence has now been revised
- Line 264: ‘Quiet a number of seroprevalence in human’ suggest removing
Response: Corrected as suggested
Conclusion
- Line 329: ‘Infects a wide rang…’
Response: Corrected as suggested
Reviewer 2 Report
The presented systematic review adresses an interesting topic (about a zoonotic disease, Q fever in Nigeria). Overall, the article is well written.
Two details deserve more attention: the time interval considered for including papers in the review and the filters utilized. Regarding the first topic: an interval of 42 or 62 years? See lines 26 and 90. Regarding the other topic, from 1,199 articles, only 15 were considered as main references. Wasn´t there an exaggeration in the filter or an error in the choice of keywords? A total of 54 references were cited in the text.
In addition to what has already been mentioned, there is a need for some specific corrections:
- on lines 26 and 27: 15 is more than 50% of 33?
- line 47: “Coxiellaceae” not in italics;
- line 58: usually asymptomatic?
- line 69: can also be infected;
- line 76: geographical distances;
- line 100: grey literature?
- lines 145-146: 48 + 121 = 169 and not 194;
- line 160: was performed in the South;
- line 166: wide range? Because of the rodents?
- line 194: hosts;
- line 200: it is necessary to explain “questing ticks”;
- line 203: transmission of...
- line 205: obtained at...
- line 212: nut-shell?
- line 247: “Gerbillinae” not in italics;
- lines 263-264: “reported quiet” in relation to seroprevalence?
- line 303: The second study...
- line 307: parts of Africa;
- line 331: vectors;
Author Response
Reviewer 2
The presented systematic review adresses an interesting topic (about a zoonotic disease, Q fever in Nigeria). Overall, the article is well written.
Two details deserve more attention: the time interval considered for including papers in the review and the filters utilized. Regarding the first topic: an interval of 42 or 62 years? See lines 26 and 90. Regarding the other topic, from 1,199 articles, only 15 were considered as main references. Wasn´t there an exaggeration in the filter or an error in the choice of keywords? A total of 54 references were cited in the text.
Response: We followed the PRISMA guidelines in conducting the study. One key element in the PRISMA protocol for conducting a systematic review is the eligibility (inclusion criteria) for any study to qualify for inclusion. Nigeria got its independence in 1960 from its colonial masters and we wanted to expressly capture that all the study that were considered for inclusion ranged from that date until when the last search was performed in 2022. However, the earliest published paper was in 1985. The large number of search item was as a result of the “keywords” used for the search but that is not a problem as the initial screening involved only looking at the title and abstract. Lastly, only relevant reference was used in writing the manuscript and thus the number of bibliography used was modest.
In addition to what has already been mentioned, there is a need for some specific corrections:
- on lines 26 and 27: 15 is more than 50% of 33?
Response: Thank you for the observation. We have now corrected it.
- line 47: “Coxiellaceae” not in italics;
Response: italics removed on ‘’Coxiellaceae’’
- line 58: usually asymptomatic?
Response: Sentence has been revised
- line 69: can also be infected;
Response: ‘also’ inserted
- line 76: geographical distances;
Response: Removed
- line 100: grey literature?
Response: this simply refers to published original literature that was obtained for inclusion from the non-conventional searches by looking up into the citations list of eligible articles that was obtained from electronic database. However, we did not obtain any of such from this study and have now deleted the grey literature from the manuscript.
- lines 145-146: 48 + 121 = 169 and not 194;
Response: Corrected to 169. Thank you for the observation.
- line 160: was performed in the South
Response: ‘’was performed’’ inserted
- line 166: wide range? Because of the rodents?
Response: The study covered several host including sheep, goats, cattle and rodents. However, we have rephrased the sentence by removing the wide range.
- line 194: hosts;
Response: ‘hosts’ corrected
- line 200: it is necessary to explain “questing ticks”;
Response: An explanatory phrase to qualify questing ticks has now been provided
- line 203: transmission of...
Response: ‘of’ inserted
- line 205: obtained at...
Response: ‘at’ inserted
- line 212: nut-shell?
Response: ‘’a’’ is removed
- line 247: “Gerbillinae” not in italics;
Response: Italics removed.
- lines 263-264: “reported quiet” in relation to seroprevalence?
Response: Removed and sentence rephrased.
- line 303: The second study...
Response: Correction effected
- line 307: parts of Africa;
Response: corrected
- line 331: vectors;
Response: correction effected
Reviewer 3 Report
In the review article," Distribution and Prevalence of Coxiella burnetii in Animals, Humans and Ticks 2 in Nigeria: A Systematic Review", Muhammed et al., provide a deeper understanding of the occurrence of C. burnetii and its connected host/vector in Nigeria. One major strength of the review article is that the authors have attempted to furnish information regarding the Q-fever's prevalence in North and South Nigeria as distinct entities. Please see my comments below:
Major:
1. In introduction, it will be helpful to comment on the inhalation infective dose for C. burnetti being less than 10 bacteria, which makes individuals such as farmers and veterinarians particularly vulnerable to contracting this disease when they come into contact with infected animals and their derivatives.
Under Diagnosis:
1. Please describe both PCR methods for detection, including conventional and real time PCR.
2. What are the predominant targeted genes for PCR detection used in Nigeria for C. burnetti?
3. Are there one or more than one targeted gene used for identification?
4. Please comment on the seroprevalence in cattles (PMID: 33075103) and sheep (PMID: 34491900). Whether any seroprevalence studies had been carried out in humans or other mammals?
Minor edits:
1. In abstract line 30, it is mentioned that "6 tick spp" were recognized but in line 154, it says "8 tick spp" which one is correct?
2. Please correct the grammatical errors.
Minor edits:
1. In line 29, please change "goat's" to "goats."
2. line 108, it should be "is the article."
3. line 204, please put "of" after the word transmission
4. Please rephrase line 227, starting the sentence with "Very interesting.." does not sound right.
Author Response
Reviewer 3
In the review article," Distribution and Prevalence of Coxiella burnetii in Animals, Humans and Ticks 2 in Nigeria: A Systematic Review", Muhammed et al., provide a deeper understanding of the occurrence of C. burnetii and its connected host/vector in Nigeria. One major strength of the review article is that the authors have attempted to furnish information regarding the Q-fever's prevalence in North and South Nigeria as distinct entities. Please see my comments below:
Major:
- In introduction, it will be helpful to comment on the inhalation infective dose for C. burnetti being less than 10 bacteria, which makes individuals such as farmers and veterinarians particularly vulnerable to contracting this disease when they come into contact with infected animals and their derivatives.
Response: Thank you for the observation. We have now provided more insight to this statement and relevant references have been cited.
Under Diagnosis:
- Please describe both PCR methods for detection, including conventional and real time PCR.
Response: Correction effected
- What are the predominant targeted genes for PCR detection used in Nigeria for C. burnetti?
Response: More insight has now been provided with respect to the targeted gene used for the detection of C. burnetii in Nigeria.
- Are there one or more than one targeted gene used for identification?
Response: Yes, more than one gene and we have provided details to this in the text.
- Please comment on the seroprevalence in cattles (PMID: 33075103)and sheep (PMID: 34491900). Whether any seroprevalence studies had been carried out in humans or other mammals?
Response: We have commented on the findings of C. burnetii screening in cattles and sheep including humans.
Minor edits:
- In abstract line 30, it is mentioned that "6 tick spp" were recognized but in line 154, it says "8 tick spp" which one is correct?
Response: 8 tick spp. correction effected
- Please correct the grammatical errors.
Response: We have attempted to correct all relevant editorial errors in the manuscript.
Comments on the Quality of English Language
Minor edits:
- In line 29, please change "goat's" to "goats."
Response: Correction effected
- line 108, it should be "is the article."
Response: correction effected
- line 204, please put "of" after the word transmission
Response: of’’ Inserted
- Please rephrase line 227, starting the sentence with "Very interesting.." does not sound right.
Response: Sentence has been rephrase.

Round 2
Reviewer 1 Report
The review is clear and well written. The topic is of interest and fulfills a need. The recommendations have been incorporated. There are three minor comments below.
Materials and Methods
- Line 90: Please clarify why January 1960 was used as the starting date for inclusion.
Response: Thank you for the comment. In systematic review, literature on a given topic over a stated period of time is synthesized to determine the pattern of disease or tredn only if they meet the eligibility criteria. Consequently, in our study, we considered all literature published on Q fever/C. burnetii infection from 1960 to date that are available on the public repository/databases.
It will help readers of your review to understand your inclusion criteria such as why the specific time frame was chosen. Is it an arbitrary time frame or was there a reason for selecting it?
- Line 98: Please clarify why milk samples were selected for inclusion over other sample types.
Response: The aim of our study was to determine the prevalence and distribution of C. burnetii in humans, animals and tick vectors in Nigeria. In other words, we were interested on a study that have screened for C. burnetii/Q fever in Nigeria irrespective of sample type. Lastly, we had no specific preference for any sample.
The response contradicts what is written in the review. Lines 107-109 state that ‘The search was restricted to articles reporting the work done on blood, and milk samples …’ – In its current form the statement reads as if the sample types are part of the inclusion criteria. Suggest editing the sentence for clarity.
-
Line 111: Please clarify what is meant by ‘Did the study capture area of study clearly?’
Response: The major focus of our study was Nigeria and any eligible study for inclusion is expected to clearly state which state/province the study was undertaken in Nigeria. This information is necessary to determine the spatial distribution of C. burnetii in the country.
SSince the phrase ‘area of study’ can mean a discipline, perhaps ‘Did the study clearly define the geographical region’ or similar would be clearer to the reader.
Author Response
Line 90: Is it an arbitrary time frame or was there a reason for selecting it?
Response: The time frame was not arbitrary. Nigeria became an independent country in 1960. Consequently, we wanted to investigate all studies that have been published from 1960 to date.
Line 98: Please clarify why milk samples were selected for inclusion over other sample types.
Response: Thank you for your comment and observation. If you read closely, the sentence is very clear. In the included studies involving domestic and peri-domestic animals, and humans, blood and milk samples only were used as template for screening either in serology (ELISA) or PCR. We concluded by saying that tick (both feeding and questing) that were screened for C. burnetii was also included. However, we have recast the sentence to provide more clarity to our thoughts.
Line 111: Please clarify what is meant by ‘Did the study capture area of study clearly?’
Response: Thank you for providing the right word “geographical region” as area of study. We would revise the sentence accordingly.